**Data Availability Statement:** All relevant data are within the paper and its Supporting information files.

**Funding:** The author(s) received no specific funding for this work.

# I-FABP is decreased in COVID-19 patients, independently of the prognosis

**Kevin Guedj** [1]*, **Mathieu Uzzan**[2], **Damien Soudan**[2], **Catherine Trichet**[3], **Antonino Nicoletti**[1], **Emmanuel Weiss**[4], **Hana Manceau** [5,6], **Alexandre Nuzzo**[2], **Olivier Corcos**[2], **Xavier Treton**[2,5], **Katell Peoc'h**[5,6]

**1** INSERM UMRS 1148 LVTS and University of Paris, Paris, France, **2** Gastroenterology Department, Beaujon Hospital, APHP, Clichy, France, **3** Haematology Department, Beaujon Hospital, APHP, Clichy, France, **4** Intensive Care Unit, Beaujon Hospital, APHP, Clichy, France, **5** CRI, INSERM UMRs 1149 and University of Paris, Paris, France, **6** Biochemistry Department, Beaujon Hospital, APHP, Clichy, France

* Kevin.guedj@gmail.com

## Abstract

### Background

Severe acute respiratory syndrome caused by the novel coronavirus (SARS-CoV-2) is frequently associated with gastrointestinal manifestations. Herein we evaluated the interest in measuring the intestinal fatty acid-binding protein (I-FABP), a biomarker of intestinal injury, in COVID-19 patients.

### Methods

Serum I-FABP was analyzed in 28 consecutive patients hospitalized for a PCR-confirmed COVID-19, in 24 hospitalized patients with non-COVID-19 pulmonary diseases, and 79 patients admitted to the emergency room for abdominal pain.

### Results

I-FABP serum concentrations were significantly lower in patients with COVID-19, as compared to patients with non-COVID-19 pulmonary diseases [70.3 pg/mL (47–167.9) vs. 161.1 pg/mL (88.98–305.2), respectively, *p = 0.008*]. I-FABP concentrations in these two populations were significantly lower than in patients with abdominal pain without COVID-19 [344.8 pg/mL (268.9–579.6)]. I-FABP was neither associated with severity nor the duration of symptoms. I-FABP was correlated with polymorphonuclear cell counts.

### Conclusions

In this pilot study, we observed a low I-FABP concentration in COVID-19 patients either with or without gastrointestinal symptoms, of which the pathophysiological mechanisms and clinical impact remain to be established. Further explorations on a larger cohort of patients will be needed to unravel the molecular mechanism of such observation, including the effects of malabsorption and/or abnormal lipid metabolism.

**Competing interests:** The authors have declared that no competing interests exist.

## Introduction

The SARS-CoV-2 infection is associated with an extensive range of symptoms, including first-line respiratory manifestations and gastrointestinal manifestations [1]. Indeed, in a study including 164 COVID-19 patients of Chinese descent, Zhang et al. evidenced gastrointestinal signs in one-third of the patients [1].

Furthermore, the SARS-CoV-2 receptor is the Angiotensin-Converting Enzyme 2 (ACE2), which is mainly expressed by enterocytes brush border of the intestinal epithelium, despite its strong expression by lung epithelial cells [2]. Therefore, identifying intestinal-derived molecules released upon SARS-CoV-2-induced damages could serve as biomarkers to monitor the pathology, particularly in long-term and/or severe patients.

We recently found that in patients with COVID-19, plasma citrulline concentration inversely correlates with systemic inflammation. Patients that presented low plasma citrulline concentrations also showed higher systemic inflammation. Moreover, low citrulline and gastrointestinal symptoms were associated with more severe diseases [3].

Indeed COVID-19 is associated with a wide range of intestinal pathologies and alterations in gut microbiota [4]. Citrulline, a non-proteinogenic amino acid, is a marker of short bowel enterocyte mass and function. This marker has also been shown to be a marker of acute mesenteric ischemia [5]. Therefore, the decrease of citrulline concentration in COVID-19 patients suggests that the infection might alter the enterocyte function, leading to a loss of function.

Thus, we are convinced that identifying circulating biomarkers of digestive complications is mandatory to improve patients' management and provide insight into the infection's pathophysiological mechanisms.

Intestinal Fatty Acid-Binging Protein (I-FABP) is a 15 kDa cytosolic protein known to be involved in the uptake and trafficking of fatty acids in the small and large intestine. The protein is mainly expressed by enterocytes located at the tips of the intestinal villi. This cytoplasmic protein is released by mature enterocytes as soon as the cell membrane integrity is compromised. I-FABP is thus the reflection of the extent of gut damage, and it is therefore used as a biomarker of mucosal injury and other diseases affecting the intestine [6].

Indeed, since enterocytes are the first affected in ischemic intestinal injuries, I-FABP has been proposed as a diagnostic marker of acute intestinal ischemia [7]. Moreover, its circulating concentration has been evidenced to present a prognostic value in patients admitted in intensive care units (ICU) in the context of several pathological conditions, including septic shock and postoperative cardiac surgery intensive care [8, 9]. To expand the possible readouts of enterocyte function and viability, we aimed at analyzing serum I-FABP in patients with COVID-19.

## Material and methods

In a single academic center (Beaujon Hospital, APHP, France), from April 14 to April 24, 2020, we prospectively enrolled twenty-eight consecutive patients hospitalized for a PCR-confirmed COVID-19 on nasopharyngeal swabs associated with pneumonia. The study was approved by the Ethics Committee of Paris-Nord Val de Seine University Hospitals, and patients gave their written informed consent.

All COVID-19 patients exhibited pneumonia. Following the ethical committee's standard (IRB 00,006,477), we collected clinical and biological data. Routine baseline clinical characteristics and biological data were collected upon admission for all patients. Patients were included either from the medical ward or directly from the Intensive Care Unit (ICU). Patients from

ICU (n = 6) were either immediately included or hospitalized in ICU within 48h after their admission at the hospital.

Blood samples were drawn at the time of COVID-19 diagnosis, collected in appropriate tubes before immediate centrifugation at 3,000 rpm for 15 min at room temperature, and subsequent storage at -80˚C until further analysis. Biological current assays were performed at the time of inclusion on blood collected on heparin (C-Reactive Protein, I-FABP, Citrulline) or EDTA (circulating polymorphonuclear cells, lymphocytes).

As control populations, we first analyzed 24 currently hospitalized patients with non-COVID-19 pulmonary diseases, including infectious disease (4%), neoplastic (8%), inflammatory or degenerative diseases (88%).

We then studied 79 patients admitted to the emergency room for abdominal pain, initially suspected of acute mesenteric ischemia, but finally discarded after CT Scan angiography. Those patients instead presented infectious (23%), inflammatory (19%), mechanical (31%), or extra-intestinal causes (27%) of abdominal pain.

I-FABP concentrations were measured in serum using the ELISA kit from Hycult Bioteck (HK40602, Uden, The Netherlands, linearity range: 47–3,000 pg/mL) following the manufacturer's instructions.

Briefly, samples and standards were incubated in 96-well microtiter plates coated with antibodies recognizing human I-FABP. The standard curve included six points ranging from 47 pg/mL to 3000 pg/mL (S1 Fig).

The biotinylated secondary antibody was then added to the wells. After washes, the streptavidin-peroxidase conjugate was added and then reacted with tetramethylbenzidine (TMB) substrate. The absorbance was then measured with a spectrophotometer at 450 nm (Infinite[®] 200 PEO, TECAN). I-FABP was assayed by batch at the end of the inclusion period. We used two quality controls (sample pool)–at the beginning and the end of the plates.

Citrulline plasma concentrations were assayed using an ultra-performance liquid chromatography-mass spectrometer (UPLC-MS; Xevo TQS, Waters[®]).

The retention time was 1.43 min on the Xevo TQS, Waters[®].

The results are presented as an average of two measurements (duplicate). If a coefficient of variation higher than 10% was observed between two measures, the sample was reanalyzed.

CRP and other common biological markers were routinely performed in the central laboratory onto Architect C8000 (Abbott) or XN-3000 (Sysmex).

For each of the continuous variables, we report the mean and standard deviation, or median and Interquartile range when indicated. Categorical variables are expressed as the number of observations and percentages. Quantitative data were analyzed with the Mann-Whitney U test. Subgroup analyses were performed with the use of the Kruskal-Wallis tests for skewed distributions. All graphs and statistical analysis were performed in Graphpad Prism 9.

## Results

Among our COVID-19 population mean age was 62 years (standard deviation -SD-: 13.94). Twenty-five percents were females. Thirteen patients presented elevated blood pressure, 14 diabetes, eight dyslipidemia. Only two exhibited abdominal pain, nausea, vomiting, 15 a recent loss of appetite, whereas 11 complained of diarrhea. As a sum, 14 out of the 28 patients exhibited almost one digestive sign. The average body mass index (BMI) was 28 (standard deviation: 3.9). The main features are summarized in Table 1. We separated patients as severe or non-severe on two criteria: hospitalization in ICU/ and or death. Five patients deceased during the hospitalization. Among patients with pulmonary diseases, the mean age was also 62 years (SD:

**Table 1. Main COVID19-patients characteristics.**

|  |  | N = 28 |
|---|---|---|
| Age (mean +/- SD) |  | 61.2 +/- 13.8 |
| Sex (female (n, %)) |  | 7 (25%) |
| Time from symptoms onset (days) |  | 14 +/- 9.88 |
| Cardiovascular risk factors | Elevated blood pressure | 13 (46.4%) |
|  | Diabetes | 14 (50.0%) |
|  | Dyslipidemia | 8 (28.5%) |
|  | Cardiovascular history | 2 (7.1%) |
| Pulmonary diseases | COPD | 2 (7.1%) |
|  | Asthma | 2 (7.1%) |
| History of IBD |  | 0 (0%) |
| Smoking |  | 1 (3.5%) |
| Medications | ARA2 | 4 (14.2%) |
|  | Steroids | 0 (0%) |
|  | NSAIDS | 0 (0%) |
| Clinical features | Fever | 18 (64.4%) |
|  | Caughing | 22 (78.6%) |
|  | Shortness of breath | 13 (46.4%) |
|  | Anosmia | 8 (28.6%) |
|  | Agueusia | 9 (40.8%) |
|  | Arthromyalgia | 9 (32.1%) |
|  | Recent loss of appetite | 14 (53.6%) |
|  | Nausea/vomiting | 3 (10.7%) |
|  | Abdominal pain | 2 (7.1%) |
|  | Diarrhea | 11 (39.3%) |
|  | Digestive symptoms | 14 (50.0%) |
| Physical characteristics | Baseline body mass index | 28 +/- 3.9 |
| Outcome | ICU | 6 (19.2%) |
|  | Orotracheal intubation | 3 (10.7%) |
|  | Death | 4 (3.9%) |

IBD: inflammatory bowel disease. ICU: Intensive Care Unit. ARA2: angiotensin 2 antireceptor. COPD: chronic obstructive pulmonary disease

12.36), with 45.8% of females, whereas in patients with abdominal pain, the mean age was 51 years years (SD:19.9), with 39% of females.

As shown in Fig 1, I-FABP serum concentrations were significantly decreased in patients with COVID-19, as compared to control patients with non-COVID-19 pulmonary diseases [Med: 70.3 pg/mL (IQ: 47–167.9) vs. 161.1 pg/mL (IQ: 88.98–305.2), respectively, *p = 0.008*].

I-FABP concentrations in these two populations were significantly lower than in patients with abdominal pain without COVID-19 [Med: 344.8 pg/mL (IQ: 268.9–579.6)].

We next assessed whether the severity of the disease was associated with serum I-FABP concentration. We found no statistically significant difference in I-FABP concentrations between patients from the ICU and those from the medical ward or between patients who died from the disease compared with patients who survived (Fig 2). No correlation was found between I-FABP concentrations and the duration of symptoms (S2 Fig).

We then analyzed the correlation of several biological markers, including biomarkers of systemic inflammation (CRP), current biological markers (circulating polymorphonuclear

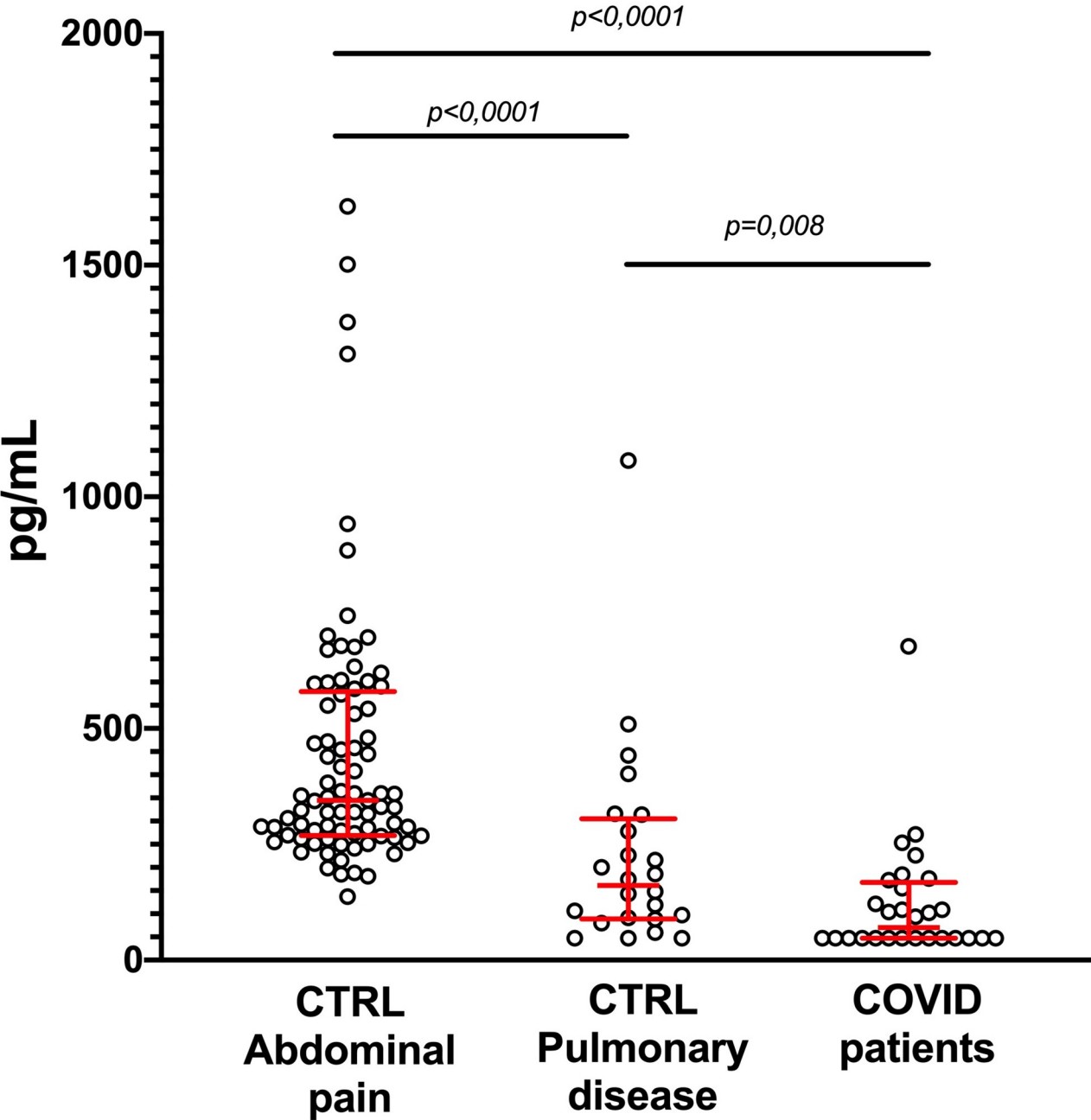

**Fig 1. Serum I-FABP concentration is decreased in COVID-19 patients.** I-FABP concentrations were measured in COVID-19 patients (n = 28) and in patients with abdominal pain or pulmonary disease (n = 79 and 24, respectively). ANOVA and Mann-Whitney tests were used for statistical comparison. CRTL: control.

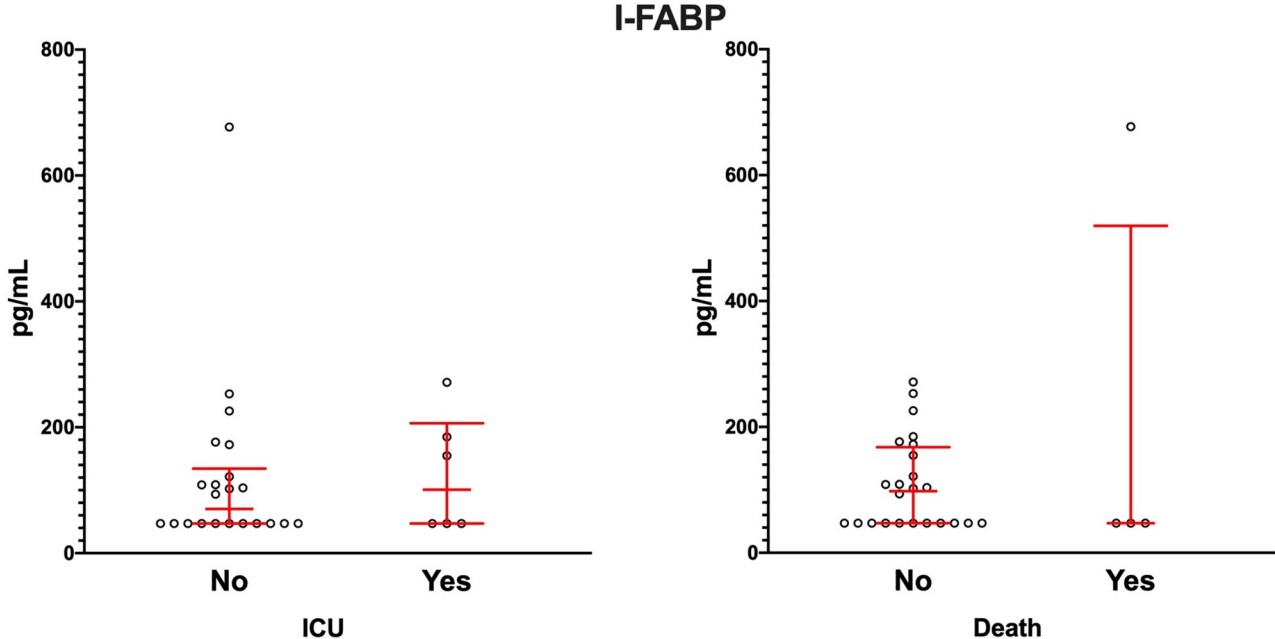

**Fig 2. Serum I-FABP concentration is not associated with severe forms of COVID-19.** I-FABP concentrations were measured in COVID-19 patients from medical ward (n = 22) and ICU (n = 6) (left panel) and patients that live (n = 24) or deceased (n = 4; right pannel). Mann-Whitney test was used for statistical comparison.

cells (PMN, Lymphocytes), and citrulline with I-FABP. The correlation table is presented in Fig 3. We found a strong, statistically significant correlation of I-FABP concentrations with the number of PMN. There was no correlation between I-FABP and citrulline or between I-FABP and systemic inflammatory biomarker (CRP) or lymphocytes.

## Discussion

In the present study, we observed that I-FABP concentrations were significantly decreased in patients with COVID-19, compared to patients with non-COVID-19 pulmonary diseases and in patients with abdominal pain without COVID-19. I-FABP is a biomarker of enterocyte injury. To date, although we did not reevaluate them in the present study, values in healthy controls for I-FABP with ELISA Hycult were previously evaluated less than 90 pg/mL, thus slightly higher than what we observed for COVID-19 patients. This result was unexpected. We also evaluated these results taking into account the potential impact of pulmonary diseases and of marked abdominal pain.

I-FABP is known to participate in fatty acid metabolism. Notably, it has been suggested that I-FABP could be involved in dietary lipid sensing and signaling, as indicated by results obtained in -IFAPB -/- mice [10]. It has been shown that lipid metabolism is modified in COVID-19 patients. Hypolipidemia is associated with the severity of COVID-19, with LDL cholesterol and Total cholesterol concentrations lower in all COVID-19 patients and HDL cholesterol [11, 12], and hypertriglyceridemia in severe patients [13]. This could partially explain our observation due to an active mechanism rather than a free passive diffusion of the marker.

On the other hand, malabsorption is a common feature in COVID-19 patients with digestive symptoms. A decrease in I-FABP may be due to fat malabsorption through enterocytes, as evidenced in coeliac disease [14]. However, possibly due to the population's small size, we did

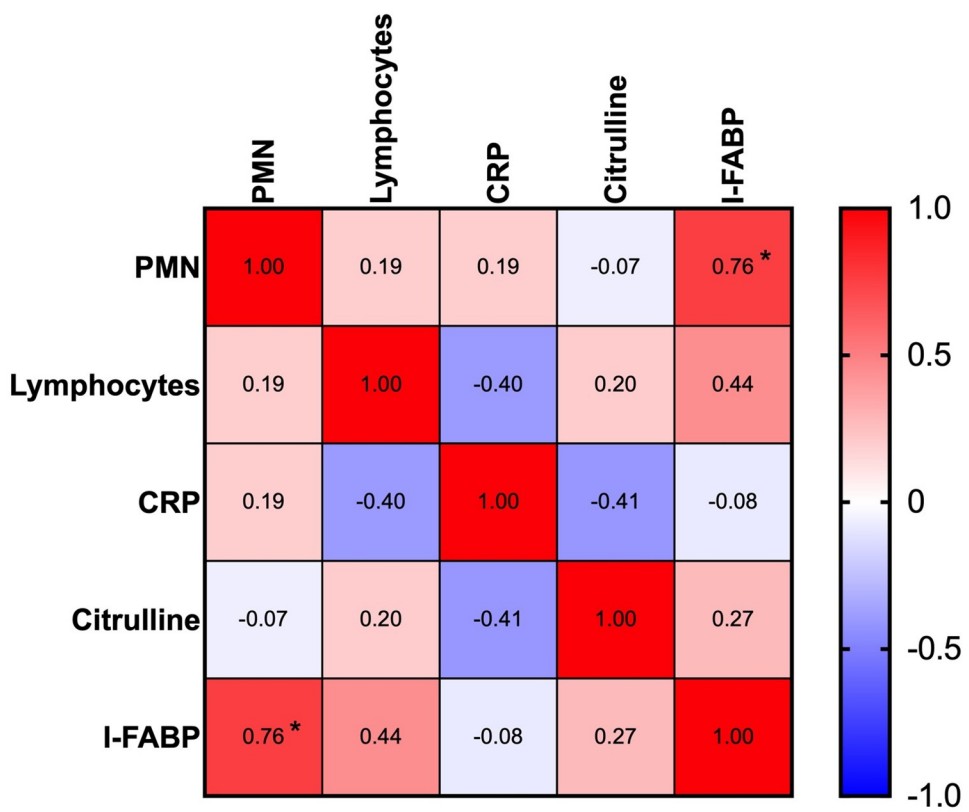

**Fig 3. Serum I-FABP correlated with the number of polymorphonuclear cells in COVID-19 patients.** Correlation table between serum concentration of I-FABP, citrulline, C reactive protein (CRP), and the number of leukocytes. Pearson correlation test was used. The correlation index is indicated. A negative sign indicates a negative correlation.

not evidence any difference in I-FABP concentrations between patients with or without gastrointestinal signs (data not shown).

Patients with systemic inflammatory response syndrome (SIRS) have increased intestinal permeability. Thus, severe COVID-19 patients are at risk of developing increased intestinal permeability with prognosis worsening. However, we did not evidence a marked difference related to the prognosis in the present study.

I-FABP has also been studied in other infective conditions, such as *Clostridioides difficile* infection [15]. Severe sepsis in ICU patients has been shown to be associated with an increase in I-FABP concentrations [16]. In both observations, I-FABP was increased in patients. But no study, to the best of our knowledge, evaluated I-FABP in acute viral infections to date.

Intriguingly, one of the four patients who died presented a much higher concentration of I-FABP compared to other COVID-19 patients. This patient also presented numerous other comorbidities, such as diabetes, hypertension and a recent loss of appetite. This observation might suggest that the increase of I-FABP in some COVID-19 patients might depend on the combination of pathological conditions that are *in fine* associated with marked alterations of the intestinal barrier, or perhaps related to bacterial diseases rather than viral diseases.

I-FABP does not correlate with the citrulline pattern we previously observed in COVID-19 patients. This observation may have several explanations, including differential evolution and half-life for both markers, a difference between the reduction of enterocyte mass and function in COVID-19 patients and the alteration of the intestinal mucosa. I-FABP is also not correlated with CRP, which reflects the inflammation climate. The correlation we observed between

PMN and I-FABP is puzzling. This positive correlation is highly significant (S1 Table). Our initial hypothesis was that I-FABP might be associated with intestinal damage in severe COVID-19 patients. The role of PMN in COVID-19 pathogenesis is still unclear. Indeed, an expanded myeloid compartment and some subtypes of PMN have been associated with the prognosis in some studies [17].

This study presents some limitations.

The number of patients we included is small, especially for the group presenting severe symptoms. I-FABP concentrations should be analyzed in larger COVID-19 patients' populations with and without gastrointestinal symptoms. More extensive studies should be undertaken to determine whether I-FABP could be used either as a prognosis factor in COVID-19 or peculiar phenotypes. Notably, it has been shown that acute mesenteric ischemia is a complication in severe forms of COVID-19 requiring hospitalization in the ICU [18]. Whether I-FABP should be used in this setting remains to be elucidated.

Additionally, COVID-19 patients frequently present diarrhea, which may affect the circulated concentration of I-FABP due to increased passage. I-FABP should also be measured in stool samples, although there is no current assay appropriate for such measurement. Finally, correlations between I-FABP with other biomarkers of COVID-19 prognosis should be informative, for instance, with Growth differentiation factor 15 (GDF-15) is associated with severe COVID-19, hypoxemia, and SarsCov2 viremia [19]. Whereas intestinal mucosal injury has been evidenced in COVID-19 [20], our first study failed to evidence I-FABP as a prognosis marker or a marker associated with gastrointestinal signs in COVID-19. However, our results are in line with those obtained by Hoel et al. [21], which showed no elevation of I-FABP in COVID-19 patients with cardiac involvement. Further investigations will be needed to decipher such observation's molecular mechanism, in line with our previous data on citrulline.

## Supporting information

**S1 Fig. Standard curve of ELISA assay.** Hycult ELISA assay was used as recommended by the manufacturer.
(TIF)

**S2 Fig. Linear regression between I-FABP concentration and symptom duration.** No significant association was seen ($R^2 = 0.06327$).
(TIF)

**S1 File.**
(XLSX)

**S1 Table. P-value of the correlation between I-FABP and current biological values.**
(DOCX)

## Author Contributions

**Conceptualization:** Kevin Guedj, Emmanuel Weiss, Olivier Corcos, Xavier Treton, Katell Peoc'h.

**Data curation:** Kevin Guedj.

**Formal analysis:** Kevin Guedj, Katell Peoc'h.

**Validation:** Catherine Trichet.

**Writing – original draft:** Kevin Guedj, Katell Peoc'h.

**Writing – review & editing:** Kevin Guedj, Mathieu Uzzan, Damien Soudan, Catherine Trichet, Antonino Nicoletti, Emmanuel Weiss, Hana Manceau, Alexandre Nuzzo, Olivier Corcos, Xavier Treton, Katell Peoc'h.

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
