## [Decision Letter · Decision Letter 0]

2 Feb 2021

PONE-D-20-39264

I-FABP is decreased in COVID-19 patients, independently of the prognosis

PLOS ONE

Dear Dr. Guedj,

Thank you for submitting your manuscript to PLOS ONE. After careful consideration, we feel that it has merit but does not fully meet PLOS ONE’s publication criteria as it currently stands. Therefore, we invite you to submit a revised version of the manuscript that addresses the points raised during the review process.

We look forward to receiving your revised manuscript.

Kind regards,

Aleksandar R. Zivkovic

Academic Editor

PLOS ONE

2.Thank you for including your ethics statement: 

" “Comité d’Evaluation de l’Ethique des projets de

Recherche Biomédicale (CEERB) Paris Nord” (Institutional Review Board -IRB 00006477- of HUPNVS, Paris 7 University, AP-HP".   

Please amend your current ethics statement to confirm that your named institutional review board or ethics committee specifically approved this study.

3.Please provide additional details regarding participant consent. In the ethics statement in the Methods and online submission information, please ensure that you have specified (1) whether consent was informed and (2) what type you obtained (for instance, written or verbal, and if verbal, how it was documented and witnessed). If your study included minors, state whether you obtained consent from parents or guardians. If the need for consent was waived by the ethics committee, please include this information.

4.We note that you have indicated that data from this study are available upon request. PLOS only allows data to be available upon request if there are legal or ethical restrictions on sharing data publicly. For more information on unacceptable data access restrictions, please see http://journals.plos.org/plosone/s/data-availability#loc-unacceptable-data-access-restrictions.

Reviewers' comments:

Reviewer's Responses to Questions

**Comments to the Author**

1. Is the manuscript technically sound, and do the data support the conclusions?

Reviewer #1: Partly

Reviewer #2: Yes

Reviewer #3: Partly

Reviewer #4: Partly

2. Has the statistical analysis been performed appropriately and rigorously? 

Reviewer #1: Yes

Reviewer #2: Yes

Reviewer #3: No

Reviewer #4: No

3. Have the authors made all data underlying the findings in their manuscript fully available?

Reviewer #1: No

Reviewer #2: Yes

Reviewer #3: No

Reviewer #4: Yes

4. Is the manuscript presented in an intelligible fashion and written in standard English?

Reviewer #1: Yes

Reviewer #2: Yes

Reviewer #3: Yes

Reviewer #4: No

5. Review Comments to the Author

Reviewer #1: Authors evaluated serum levels of the intestinal fatty acid-binding protein (I-FABP), a biomarker of intestinal injury, in COVID-19 patients. It’s an important study considering the digestive system involvement in COVID-19, however, it presents a negative result against the authors’ assumed hypothesis. Authors showed (I-FABP) is significantly decreased in COVID-19 patients for which they are not able to explain the patho-physiological mechanisms and the clinical significance.

Major concerns:

1. Materials and method section is very short, more elaboration is necessary.

2. Materials and method: Authors write they correlated several biological markers, including biomarkers of systemic inflammation, current biochemical markers, and citrulline with I-FABP in COVID-19 patients. The descriptions of these markers are missing in the paper. Please provide complete list, describe the methods used in their estimation, and detailed results of the study for these markers. Why these markers were not studied for the control groups?

3. Materials and method: Include standard curve for the ELISA estimation. Also include details for how many times ELISA was performed and how many multiples were used for each sample.

4. Materials and method: Details for statistical analysis is completely missing, please include necessary details.

5. Results: As it’s not a brief communication, present results separately from the discussion. All patients with COVID-19 don’t have digestive symptoms, analyzing the results separately for those with digestive symptoms may be more informative.

6. Discussion: Even if authors are not able to explain pathophysiological basis of decreased expression of I-FABP, they can present some assumptions based on the literature evidence, that will add clinical significance to this study. Malabsorption is a common feature in COVID-19 in COVID-19 patients with digestive symptoms. It’s possible that a decrease in I-FABP may be due to fat malabsorption through enterocytes. Ref: Decreased expression of Intestinal I- and L-FABP levels in rare human genetic lipid malabsorption syndromes. https://link.springer.com/article/10.1007/s00418-007-0302-x

Hypolipidemia is associated with the severity

of COVID-19: https://pubmed.ncbi.nlm.nih.gov/32430154/

7. Discussion: Authors write that “I-FABP does not follow the citrulline pattern in COVID-19, which suggests that the reduction of enterocyte mass and function in COVID-19 patients is not associated with an alteration of the intestinal mucosa.” This is a far reaching interpretation of their results. In contrast of their claim, current evidence suggests clear cut intestinal mucosal injury in COVID-19 (Ref: COVID-19 and the Digestive System:https://www.ncbi.nlm.nih.gov/pmc/articles/PMC7273952/).

8. Discussion: Authors should avoid reaching this conclusion with their limited investigation: “we can conclude that the interest of I-FABP to highlight the risk of intestinal complications in these patients is however unlikely, according to our results.” Instead they should conclude with what were the main outcomes of this study.

9. Discussion: Although, finding of this study is important, this reviewer believes that a single ELISA based assay is not sufficient to validate results of this study. Also sample size of the study is small, especially for the test group presenting patients with severe symptoms. Please include a section of limitations and provide future directions how the results can be further validated.

Reviewer #2: Plasma level of I- FABP is used as a validated prognostic marker of gut damage for IBDs, acute ischemia and HIV. (Ref Isnard S. Plasma Levels of C-Type Lectin REG3α and Gut Damage in People With Human Immunodeficiency Virus. J Infect Dis. 2020 Jan 1;221(1):110-121). As ACE is strongly expressed in gut investigators assessed serum I-FABP in small groups of patients with COVID-19.

Contra intuitively COVID patients had only half mean value when compared to controls. They did not observed correlation between I-FABP and citrulline or CRP). IFABP is known to participate in fatty acid metabolism, and GDF15 is known to decreased HDL and LDL cholesterol and hypertriglyceridemia in severe COVID patients.

Myhre PL. Growth Differentiation Factor 15 Provides Prognostic Information Superior to Established Cardiovascular and Inflammatory Biomarkers in Unselected Patients Hospitalized With COVID-19. Circulation. 2020 Dec;142(22):2128-2137.

Lymphocytes ratio on PMN or neutrophil should also be presented as one of easy-to-get prognosis for COVID for I FABP, CRP and citrulline correlation.

Assessment of GDF15 will be of interest.

Interesting hypothesis generating observation.

Reviewer #3: The idea is quite interesting, however there are some important methodological issues.

There is no information about statistical analysis.

I-FABP concentration should be analyzed in COVID-19 patients with and without gastrointestinal symptoms.

Additionally, majority of COVID-19 patients present diarrhea, which may affect concentration of I-FABP due to increased passage. Then I-FABP should be also measured in stool samples.

The discussion is too short.

Reviewer #4: The authors presented a potentially interesting study with the aim of evaluating the value of circulating I-FABP in patients with COVID-19, as expression of intestinal damage. Although interesting, since a not negligible rate of patients presents with intestinal symptoms and eventually might have intestinal damage even in the absence of intestinal symptoms, the manuscript lacks of fundamental aspects that compromise its entire value.

Overall, the manuscript is poor and sometimes lacks of clarity.

The methods are poorly represented. The control groups lack crucial information.

The authors considered their study as a research article. However, it is more appropriate to consider the paper as a research note or a short communication. In fact, the paper is very short, only 8 references are presented and overall the manuscript is quite poor.

In detail:

Introduction

- In the introduction, authors presented some results on plasma levels of citrulline in COVID-19 and acute intestinal ischemia. However, the sentence is not very clear. Therefore, I would suggest authors to re-write the sentence in order to better explain the behaviour of this marker in the mentioned conditions.

- I-FABP has been also studied in other infective conditions, such as Clostridioides difficile (see Oliva et al, Open Forum Infect Dis. 2019 Dec 3;7(1):ofz507)

Methods

- “Time of study inclusion” means time of COVID-19 diagnosis?

- How was diagnosis of COVID-19 made? PCR-based methods on which sample?

- Were patients with COVID-19 with or without pneumonia? Please include in the method.

- How authors assessed COVID-19 disease severity? It must be stated in the method section along with the appropriate reference.

- “As control populations, we analyzed 24 currently hospitalized patients with non-COVID-19 pulmonary diseases.”: Why did the author chose non-COVID pulmonary diseases as control population? Was the decision based on the possibility of the gut-lung axis? Were these pulmonary diseases infective conditions (such as pneumonia) or non infective? There is no mention in the method section. Results might differ according to these conditions.

- How the authors choose the controls? How were the levels of I-FABP in healthy controls? Healthy subjects might be included as a control group in order to evaluate whether COVID-19 patients have values similar to those observed in healthy controls.

- I believe that controls group should have been matched with study population: for instance, age and sex might influence the value of I-FABP. Values might be very different if a patient with COVID-19 is a young female and control patient with pulmonary condition is a 90-year old man.

- How were the sample stored in the period comprised between sample collection and analysis? Please explain.

- I suppose that patients were divided into ICU and non-ICU and died/survived groups, respectively. However, no mention in the method section is present. Would the author consider ICU-group as patients admitted to ICU during hospitalization or directly from the Emergency Room Department? Please specify.

- Statistical analyses description are absent in the method.

Results and discussion

- Overall, how many patients presented with gastrointestinal disease? Authors described single symptoms, however, it could be interesting to know the total number of patients with gastrointestinal involvement.

- Overall, results (i.e. comorbidities) might be better presented and described.

- How many patients had COVID-19 associated pneumonia?

- It could be interesting to know the duration of symptoms before I-FABP collection. Is there a correlation between I-FABP levels and duration of symptoms?

- Authors should better describe the patient who died with high levels of I-FAB, preferably in the discussion. What does it mean anorexia? How diagnosis of anorexia was made?

- “This observation might suggest that the increase of I-FABP in COVID-19 patients might depend on the combination of pathological conditions that are in fine associated with increased alterations of the intestinal barrier”: this could be partly true. However, the authors should better explain and argue this sentence. Accordingly, authors should insert appropriate references.

- “We then analyzed the correlation of several biological markers, including biomarkers of systemic inflammation, current biochemical markers, and citrulline with I-FABP”. This part belongs to the methods section. Which biological markers were analyzed? Please specify. How was analyzed citrulline? Please specify in the method.

- Is the correlation between I-FABP and PMN positive or negative? This is not clear from the text; therefore, reading the discussion regarding this finding is quite difficult.

- The authors hypothesis was that in COVID-19 patients an intestinal damage is present, leading to the expectation of high I-FAB levels. However, they found lower level of I-FABP. How did the authors motivate and explain their findings? With this regard, the discussion is very poor and does not support the study results.

- “While the link of I-FABP levels and lipid metabolism should be further investigated in COVID-19 patients, we can conclude that the interest of I-FABP to highlight the risk of intestinal complications in these patients is however unlikely, according to our results.” This sentence is not clear. Was the aim of the authors to evaluate the possible intestinal damage in COVID-19 patients or the evaluation of intestinal complication in COVID-19 patients? If the aim, as I understood, is to evaluate the possible intestinal damage in COVID-19 patients, this sentence should be re-written The conclusions are not supported by the results of the study. Authors should re-write the last sentence of the study.

- No mention on the number of death or ICU admission is present in the result section.

- A table with study population characteristics would be appropriate

Figures

- Figure1. The number of patients with abdominal pain is 80; in the text is 79. Please correct.

- Figure3. It seems that only the coefficient is presented. Authors should also include the p value.

6. PLOS authors have the option to publish the peer review history of their article (what does this mean?). If published, this will include your full peer review and any attached files.

Reviewer #1: **Yes: **Ashutosh Kumar

Reviewer #2: **Yes: **Jean-Pierre Routy

Reviewer #3: No

Reviewer #4: No

---

## [Author Response · Author response to Decision Letter 0]

23 Mar 2021

Dear Editor,

On behalf of all the co-authors, I would like to thank you and the reviewers for the valuable comments and the significant heed in improving our manuscript.

You will find detailed answers to all of the comments, including the comments made by the editor and those made by the reviewers. Modifications suggested by the editor and/or reviewers appear in blue in the manuscript. We hope you will find the manuscript is now suitable for publication.

No ethical or legal restrictions on the diffusion of data after anonymization. We provide the corresponding files.

Answer to editor comments

" “Comité d’Evaluation de l’Ethique des projets de Recherche Biomédicale (CEERB) Paris Nord” (Institutional Review Board -IRB 00006477- of HUPNVS, Paris 7 University, AP-HP".

Please amend your current ethics statement to confirm that your named institutional review board or ethics committee specifically approved this study.

For additional information about PLOS ONE ethical requirements for human subjects research, please refer to http://journals.plos.org/plosone/s/submission- guidelines#loc-human-subjects-research.

We have now introduced this statement in the text. The project was approved by the CEERB.

The study was approved by the Ethics Committee of Paris-Nord Val de Seine University Hospitals.

3.Please provide additional details regarding participant consent. In the ethics statement in the Methods and online submission information, please ensure that you have specified (1) whether consent was informed and (2) what type you obtained (for instance, written or verbal, and if verbal, how it was documented and witnessed). If your study included minors, state whether you obtained consent from parents or guardians. If the need for consent was waived by the ethics committee, please include this information.

All patients gave their informed consent for participating in the study. All patients get a preliminary written information on the protocol. These points are now precsised in the text.

4.We note that you have indicated that data from this study are available upon request. PLOS only allows data to be available upon request if there are legal or ethical restrictions on sharing data publicly. For more information on unacceptable data access restrictions, please see http://journals.plos.org/plosone/s/data-availability#loc-unacceptable-data-access-restrictions.

We have identified no ethical or legal restrictions on the diffusion of data, after anonymization. However, IFABP values in controls with abdominal pain are currently submitted as controls in another study, and thus, we provided only some minimal data.

b) If there are no restrictions, please upload the minimal anonymized data set necessary to replicate your study findings as either Supporting Information files or to a stable, public repository and provide us with the relevant URLs, DOIs, or accession numbers. For a list of acceptable repositories, please

see http://journals.plos.org/plosone/s/data-availability#loc-recommended- repositories.

We have now added these data. However, those data are presented at this time in french...Please precise if an english version is awaited.

 Answer to reviewers

Reviewer #1: Authors evaluated serum levels of the intestinal fatty acid-binding protein (I-FABP), a biomarker of intestinal injury, in COVID-19 patients. It's an important study considering the digestive system involvement in COVID-19, however, it presents a negative result against the authors' assumed hypothesis. Authors showed (I-FABP) is significantly decreased in COVID-19 patients for which they are not able to explain the patho-physiological mechanisms and the clinical significance. Major concerns:

1. Materials and method section is very short, more elaboration is necessary.

We have added a more detailed description of the material and methods

2. Materials and method: Authors write they correlated several biological markers, including biomarkers of systemic inflammation, current biochemical markers, and citrulline with I-FABP in COVID-19 patients. The descriptions of these markers are missing in the paper. Please provide complete list, describe the methods used in their estimation, and detailed results of the study for these markers. Why these markers were not studied for the control groups?

We aimed to study I-FABP and designed the study to perform so. In comparison, COVID19 patients were extensively explored. We previously published data on citrulline in a previous manuscript, and

3. Materials and method: Include standard curve for the ELISA estimation. Also include details for how many times ELISA was performed and how many multiples were used for each sample.

We have now added the curve in supplementary data, and completed the text.

4. Materials and method: Details for statistical analysis is completely missing, please include necessary details.

 We have added some missing information concerning statistical analysis.

5. Results: As it's not a brief communication, present results separately from the discussion. All patients with COVID-19 don't have digestive symptoms, analyzing the results separately for those with digestive symptoms may be more informative.

We have now presented the results separately from the discussion and developed this latter one.

6. Discussion: Even if authors are not able to explain pathophysiological basis of decreased expression of I-FABP, they can present some assumptions based on the literature evidence, that will add clinical significance to this study. Malabsorption is a common feature in COVID-19 in COVID-19 patients with digestive symptoms. It's possible that a decrease in I-FABP may be due to fat malabsorption through enterocytes. Ref: Decreased expression of Intestinal I- and L-FABP levels in rare human genetic lipid malabsorption

syndromes. https://link.springer.com/article/10.1007/s00418-007-0302-x Hypolipidemia is associated with the severity

of COVID-19: https://pubmed.ncbi.nlm.nih.gov/32430154/

We have now included these two points in the discussion section.

7. Discussion: Authors write that "I-FABP does not follow the citrulline pattern in COVID-19, which suggests that the reduction of enterocyte mass and function in COVID-19 patients is not associated with an alteration of the intestinal mucosa." This is a far reaching interpretation of their results. In contrast of their claim, current evidence suggests clear cut intestinal mucosal injury in COVID-19 (Ref: COVID-19 and the Digestive System:https://www.ncbi.nlm.nih.gov/pmc/articles/PMC7273952/). We have now included this evidence in the discussion section.

8. Discussion: Authors should avoid reaching this conclusion with their limited investigation: "we can conclude that the interest of I-FABP to highlight the risk of intestinal complications in these patients is however unlikely, according to our results." Instead they should conclude with what were the main outcomes of this study.

We have now changed the discussion.

9. Discussion: Although, finding of this study is important, this reviewer believes that a single ELISA based assay is not sufficient to validate results of this study. Also sample size of the study is small, especially for the test group presenting patients with severe symptoms. Please include a section of limitations and provide future directions how the results can be further validated.

We have included a section on limitations and gave some perspectives.

Reviewer #2: Plasma level of I- FABP is used as a validated prognostic marker of gut damage for IBDs, acute ischemia and HIV. (Ref Isnard S. Plasma Levels of C-Type Lectin REG3α and Gut Damage in People With Human Immunodeficiency Virus. J Infect Dis. 2020 January 1;221(1):110-121). As ACE is strongly expressed in gut investigators assessed serum I-FABP in small groups of patients with COVID-19. Contra intuitively COVID patients had only half mean value when compared to controls. They did not observed correlation between I-FABP and citrulline or CRP). IFABP is known to participate in fatty acid metabolism, and GDF15 is known to

 decreased HDL and LDL cholesterol and hypertriglyceridemia in severe COVID patients.

Myhre PL. Growth Differentiation Factor 15 Provides Prognostic Information Superior to Established Cardiovascular and Inflammatory Biomarkers in Unselected Patients Hospitalized With COVID-19. Circulation. 2020 Dec;142(22):2128-2137. Lymphocytes ratio on PMN or neutrophil should also be presented as one of easy-to- get prognosis for COVID for I FABP, CRP and citrulline correlation.

Assessment of GDF15 will be of interest.

Interesting hypothesis generating observation.

We explored whether the Lymphocytes ratio on PMN ratio was correlated with I-Fabp or different within the different populations. It was not significant, and we did not include it in the final version. We have added the GDF15 perspective in the perspective section.

Reviewer #3: The idea is quite interesting, however, there are some important methodological issues.

There is no information about statistical analysis.

It has been now added in the method section.

I-FABP concentration should be analyzed in COVID-19 patients with and without gastrointestinal symptoms. Additionally, majority of COVID-19 patients present diarrhea, which may affect concentration of I-FABP due to increased passage. Then I-FABP should be also measured in stool samples.

These perspectives have been included at the end of the discussion.

The discussion is too short.

We have further discussed the results, including the hypothesis for our observation.

Reviewer #4: The authors presented a potentially interesting study with the aim of evaluating the value of circulating I-FABP in patients with COVID-19, as expression of intestinal damage. Although interesting, since a not negligible rate of patients presents with intestinal symptoms and eventually might have intestinal damage even in the absence of intestinal symptoms, the manuscript lacks of fundamental aspects that compromise its entire value.

Overall, the manuscript is poor and sometimes lacks of clarity.

The methods are poorly represented. The control groups lack crucial information. The authors considered their study as a research article. However, it is more appropriate to consider the paper as a research note or a short communication. In fact, the paper is very short, only 8 references are presented and overall the manuscript is quite poor.

In detail:

Introduction

- In the introduction, authors presented some results on plasma levels of citrulline in COVID-19 and acute intestinal ischemia. However, the sentence is not very clear. Therefore, I would suggest authors to rewrite the sentence in order to better explain the behaviour of this marker in the mentioned conditions.

We have rewritten this sentence as follow :

We recently found that in patients with COVID-19, plasma citrulline concentration inversely correlates with systemic inflammation: Patients that presented low plasma

 citrulline concentrations also showed higher systemic inflammation. Moreover, low citrulline and gastrointestinal symptoms were associated with more severe diseases.

- I-FABP has also been studied in other infective conditions, such as Clostridioides difficile (see Oliva et al, Open Forum Infect Dis. 2019 December 3;7(1):ofz507) We have introduced this idea and this reference in the discussion

Methods

- "Time of study inclusion" means time of COVID-19 diagnosis?

Yes, and blood samples were drawn at the time of COVID-19 diagnosis. We have now precise this point in the discussion.

- How was diagnosis of COVID-19 made? PCR-based methods on which sample? We prospectively enrolled twenty-eight consecutive patients hospitalized for a PCR- confirmed COVID-19 on nasopharyngeal swabs.

We have now precise this point.

- Were patients with COVID-19 with or without pneumonia? Please include in the method.

All patients exhibited pneumonia. This notion has been introduced in the text.

- How authors assessed COVID-19 disease severity? It must be stated in the method section along with the appropriate reference.

We separate patients as severe or non-severe on two criteria: hospitalization in an ICU/ and or death. This classification is now introduced in the text.

- "As control populations, we analyzed 24 currently hospitalized patients with non- COVID-19 pulmonary diseases.":

Why did the author chose non-COVID pulmonary diseases as control population? Was the decision based on the possibility of the gut-lung axis?

Were these pulmonary diseases infective conditions (such as pneumonia) or non infective? There is no mention in the method section. Results might differ according to these conditions.

We first compared our results with patients with abdominal pain of various origins. As COVID19 patients' results were lower, we wanted to know whether the pulmonary disease could be associated with a significant decrease in I-FABP. Still, we did not found this data in the literature. We have now precise the type of pulmonary disease.

- How the authors choose the controls? How were the levels of I-FABP in healthy controls? Healthy subjects might be included as a control group to evaluate whether COVID-19 patients have values similar to those observed in healthy controls.

We have no healthy subject population. We did not get any authorization to include healthy subjects in the present study. According to the literature, normal values for IFABP are less than 90 pg/mL using the Hycult ELISA (Guzel et al, Surg Today 2014). Thus these data are in the range of those observed in COVID-19 patients.

- I believe that controls group should have been matched with study population: for instance, age and sex might influence the value of I-FABP. Values might be very different if a patient with COVID-19 is a young female and control patient with pulmonary condition is a 90-year old man.

We initially did not match controls and patients according to age and sex. However, the mean age and the sex ratio are now precised in both populations.If necessary we can now perform the matching according to the age and sex.

 - How were the sample stored in the period comprised between sample collection and analysis? Please explain.

Blood samples were drawn at the time of COVID-19 diagnosis, collected in appropriate tubes before immediate centrifugation at 3,000 rpm for 15 min at room temperature for sera, and subsequent storage at -80°C until further analysis.

- I suppose that patients were divided into ICU and non-ICU and died/survived groups, respectively. However, no mention in the method section is present. Would the author consider ICU-group as patients admitted to ICU during hospitalization or directly from the Emergency Room Department? Please specify.

We have now precise the selection. ICU-group were patients. Regarding the ICU patients, there were both

- Statistical analyses description are absent in the method.

We have added this information.

Results and discussion

- Overall, how many patients presented with gastrointestinal disease? Authors described single symptoms, however, it could be interesting to know the total number of patients with gastrointestinal involvement.

Among the whole COVID19 population, fourteen patients exhibited gastrointestinal symptoms. We have now precise this point in the text.

- Overall, results (i.e., comorbidities) might be better presented and described.

We have now included a table to present variables on patients.

- How many patients had COVID-19 associated pneumonia?

All patients exhibited pneumonia. This point has been added to the text.

- It could be interesting to know the duration of symptoms before I-FABP collection. Is there a correlation between I-FABP levels and duration of symptoms?

We found no correlation between I-FABP concentration and duration of symptoms. We have now added these data as supplemental Figure 2.

- Authors should better describe the patient who died with high levels of I-FAB, preferably in the discussion. What does it mean anorexia? How diagnosis of anorexia was made?

We have corrected in the text: it was, in fact, a recent loss of appetite and not anorexia. So we have modified the text, since I is not a pathological preexisting condition.

- "This observation might suggest that the increase of I-FABP in COVID-19 patients might depend on the combination of pathological conditions that are in fine associated with increased alterations of the intestinal barrier": this could be partly true. However, the authors should better explain and argue this sentence. Accordingly, authors should insert appropriate references.

- "We then analyzed the correlation of several biological markers, including biomarkers of systemic inflammation, current biochemical markers, and citrulline with

 I-FABP". This part belongs to the methods section. Which biological markers were analyzed? Please specify. How was analyzed citrulline? Please specify in the method.

These has been added in the method section.

- Is the correlation between I-FABP and PMN positive or negative? This is not clear from the text; therefore, reading the discussion regarding this finding is quite difficult.` The correlation is positive, and we have included the notion in the text.

- The authors hypothesis was that in COVID-19 patients an intestinal damage is present, leading to the expectation of high I-FAB levels. However, they found lower level of I-FABP. How did the authors motivate and explain their findings? With this regard, the discussion is very poor and does not support the study results.

We have rewritten the discussion. Our hypothesis is that an increase of I-FABP is associated with some peculiar phenotypes of COVID-19, possibly those associated with bacterial infections.

- "While the link of I-FABP levels and lipid metabolism should be further investigated in COVID-19 patients, we can conclude that the interest of I-FABP to highlight the risk of intestinal complications in these patients is however unlikely, according to our results." This sentence is not clear. Was the aim of the authors to evaluate the possible intestinal damage in COVID-19 patients or the evaluation of intestinal complication in COVID-19 patients? If the aim, as I understood, is to evaluate the possible intestinal damage in COVID-19 patients, this sentence should be rewritten The conclusions are not supported by the results of the study.

Authors should rewrite the last sentence of the study.

We have completely change the discussion and conclusion. We really think these modifications improve the lwhole manuscript.

- No mention on the number of death or ICU admission is present in the result section.

We have added this information both in table 1 and figure legends.

- A table with study population characteristics would be appropriate

We have included the table with patients characteristic.

Figures

- Figure 1. The number of patients with abdominal pain is 80; the text is 79. Please correct.

We have corrected this mistake, 79 patients were included.

- Figure 3. It seems that only the coefficient is presented. Authors should also include the p-value.

We have added the value in supplemental table 1.

---

## [Editor Report · Decision Letter 1]

25 Mar 2021

I-FABP is decreased in COVID-19 patients, independently of the prognosis

PONE-D-20-39264R1

Dear Dr. Guedj,

We’re pleased to inform you that your manuscript has been judged scientifically suitable for publication and will be formally accepted for publication once it meets all outstanding technical requirements.

Kind regards,

Aleksandar R. Zivkovic

Academic Editor

PLOS ONE

---

## [Editor Report · Acceptance letter]

6 Apr 2021

PONE-D-20-39264R1 

I-FABP is decreased in COVID-19 patients, independently of the prognosis 

Dear Dr. Guedj:

I'm pleased to inform you that your manuscript has been deemed suitable for publication in PLOS ONE. Congratulations! Your manuscript is now with our production department. 

Kind regards, 

on behalf of

Dr. Aleksandar R. Zivkovic 

Academic Editor

PLOS ONE